# Integrative Taxonomy Reveals Two New *Trichoderma* Species and a First Mexican Record from Coffee Soils in Veracruz

**DOI:** 10.3390/jof11120856

**Published:** 2025-12-01

**Authors:** Rosa María Arias Mota, Rosario Gregorio Cipriano, Alondra Guadalupe Martínez Santos, Gabriela Heredia Abarca

**Affiliations:** Instituto de Ecología A.C., Red de Biodiversidad y Sistemática, Carretera Antigua a Coatepec No. 351, Col. Congregación El Haya, Xalapa 91073, Veracruz, Mexicoamartinezsantos88@gmail.com (A.G.M.S.);

**Keywords:** coffee rhizosphere, Hypocreales, new species, multilocus phylogeny, soil-inhabiting, taxonomy

## Abstract

*Trichoderma* species are globally distributed fungi with remarkable biotechnological relevance. In this study, we describe two new species, *T. jilotepecense* and *T. sanisidroense*, and report *T. endophyticum* as the first record for Mexico. All isolates were obtained from soils of coffee agroecosystems in Veracruz. Species identification was based on the integration of cultural and micromorphological characteristics (PDA, SNA, CMD; 25–35 °C) with multilocus phylogenetic analyses using the ITS, tef1, and rpb2 markers. The concatenated dataset provided strong support for species delimitation and clarified phylogenetic relationships within the Harzianum and Virens clades.

## 1. Introduction

The genus *Trichoderma* (Ascomycota, Hypocreales, Hypocreaceae), established by Persoon in 1794 and typified by *T. viride*, currently comprises more than 550 described and recognized species worldwide [1,2,3,4,5,6]. Members of this genus are cosmopolitan, colonizing a wide range of substrates and ecosystems across tropical and temperate regions [7]. As part of the soil fungal community, they play pivotal ecological roles as decomposers of organic matter and as participants in various biotic interactions, including mycoparasitism and symbiosis with plants and insects. These relationships confer *Trichoderma* species with remarkable potential as biological control agents, plant growth promoters, and enhancers of plant immunity [8,9,10]. Their agricultural importance has been increasingly acknowledged for improving crop productivity and sustainability, particularly through their application as biofertilizers and phosphate-solubilizing microorganisms [11,12,13].

Taxonomic identification within *Trichoderma* remains challenging due to the high degree of morphological similarity among species, which typically possess branched conidiophores bearing cylindrical to subglobose phialides and ellipsoidal to globose conidia. Consequently, molecular phylogenetic analyses have become indispensable for reliable species delimitation [1,14,15]. The sexual morph of *Trichoderma*, historically known as *Hypocrea*, is characterized by the formation of perithecial stromata with ostiolate perithecia containing cylindrical asci and bicellular ascospores that may fragment into part-ascospores. Although many species are known only from their asexual state, the teleomorph remains taxonomically relevant, providing diagnostic features, such as ascospore and stroma morphology, that complement molecular delimitation and offer a more complete evolutionary context for the genus. Among the molecular markers available, the internal transcribed spacer (ITS) region, the second largest subunit of RNA polymerase II (rpb2), and the translation elongation factor 1-alpha (tef1) are considered the most informative and widely used for species identification [5,16,17,18,19,20,21,22]. Other loci such as calmodulin (cal), actin (act), ATP citrate lyase (acl1), and endochitinase (chi18-5) have also been explored, though they generally provide lower phylogenetic resolution or limited taxa coverage [10,16,17].

In Mexico, 57 *Trichoderma* species have been reported to date, including the widely distributed *T. spirale* and *T. harzianum*. The state of Veracruz harbors at least 18 species, most of them associated with coffee rhizospheres [23,24]. Coffee plantations in this region represent tropical agroecosystems characterized by stable microclimatic conditions and organic matter-rich soils that favor the establishment and diversification of saprotrophic and mycoparasitic fungi such as *Trichoderma*. However, despite its ecological and biotechnological importance, the diversity of this genus in Mexican coffee systems remains insufficiently documented.

Integrative taxonomic approaches that combine multilocus phylogenetic inference with detailed morphological characterization are therefore essential to resolve species boundaries and uncover hidden fungal diversity in tropical ecosystems. Beyond their systematic value, such studies provide critical insights into ecosystem functioning, biogeographic patterns, and the ecological roles of fungi, thus supporting biodiversity conservation and sustainable agricultural strategies. In this study, we describe two new *Trichoderma* species and report a new record from coffee plantation soils in Veracruz, Mexico. Species identification was achieved through an integrative analysis of morphological and molecular data based on three genetic markers (ITS, rpb2, and tef1), contributing to a deeper understanding of *Trichoderma* diversity and its ecological significance in tropical environments.

## 2. Materials and Methods

### 2.1. Selection of Fungi

Strains were isolated in August 2021 from the rhizosphere of *Coffea arabica* var. Costa Rica using the soil particle filtration technique. Sampling was conducted in three coffee farms under traditional polyculture management in Jilotepec, Veracruz (Figure 1). Los Bambus, 19°36′42.74″ N, 96°60′16.01″ W; La Barranca, 19°36′38.07″ N, 96°55′40.57″ W and San Isidro, 19°36′12.15″ N, 96°54′44.91″ W. In each farm, five independent sampling points were established at 50 m intervals. At each point, approximately 250 g of rhizospheric soil was collected from a depth of 15 cm, air-dried, and stored in paper bags until further processing. The fungi were isolated using soil particle filtration technique. From a total of 225 fungal isolates obtained [25], seven representative *Trichoderma* strains were selected for detailed morphological and molecular analyses. The isolates were transferred to tubes containing potato dextrose agar (PDA medium (DIBICO S.A. DE C.V., Cuautitlán Izcalli México) and subsequently lyophilized using a Benchtop Freeze Dryer (Cat No. 77500-00, LABCONCO Corporation, Kansas, MO, USA). All *Trichoderma* strains were preserved at 4 °C in a refrigerator and at –80 °C in an ultra-freezer (Mod. 8809 7400A, Thermo scientific Inc., Waltham, MA, USA). For experimental procedures, the strains were reactivated on PDA plates prior to use.

### 2.2. Growth Rate Determination and Morphology

Morphological observations and growth rate assays were performed following the procedures described by Zhao et al. [22]. Each isolate was transferred to 9 cm Petri dishes containing three culture media: potato dextrose agar (PDA; 40 g potato dextrose per 1 L distilled water), corn meal dextrose agar (CMD; 40 g corn meal, 20 g glucose, 20 g agar per 1 L distilled water), and synthetic low-nutrient agar (SNA; 1 g KH_2_PO_4_, 1 g KNO_3_, 0.5 g MgSO_4_, 0.5 g KCl, 0.2 g glucose, 0.2 g sucrose, 20 g agar per 1 L distilled water). Plates were incubated at 25 °C, 30 °C and 35 °C under a 12 h light/12 h dark photoperiod. Inocula were placed 10–15 mm from the plate edge, and colony diameters were measured after 72 h of incubation.

Micromorphological features were examined using 3-day-old microcultures grown on PDA following the method of Chaverri et al. [17]. Permanent preparations were mounted in a mixture of lactic acid (J.T. Baker, 0194-01, Phillipsburg, PA, USA) and polyvinyl alcohol (Fagalab, 2529-500, Sinaloa, Mocorito, Mexico). For each isolate, at least 20 measurements were taken per structure (phialides, conidia, and chlamydospores) under a Nikon Eclipse Ni microscope (Nikon Corp., Tokyo, Japan) equipped with Nomarski differential interference contrast optics. Photomicrographs were captured using a Nikon Sight DS-Fi2/DS-Fi1/DS-Vi1 digital camera system. All *Trichoderma* cultures were deposited in the Fungal Culture Collection of the Instituto de Ecologia, A.C. (INECOL), Xalapa, Veracruz, Mexico.

### 2.3. DNA Extraction, PCR Amplification and Sequencing

Genomic DNA was extracted from actively growing mycelium of the seven *Trichoderma* strains cultured on PDA plates and incubated at 25 °C for three days. Mycelium was collected by gently scraping the colony surface with a sterile needle and transferred into 1.5 mL microcentrifuge tubes containing 300 µL of extraction buffer. DNA extraction was performed using the Wizard^®^ Genomic DNA Purification Kit (Promega, Madison, WI, USA) according to the manufacturer’s instructions. Three loci were amplified by Polymerase Chain Reaction (PCR): the internal transcribed spacer (ITS1–5.8S–ITS2) region, a fragment of the second largest subunit of RNA polymerase II (rpb2), and a fragment of the translation elongation factor 1-alpha (tef1) gene. The following primer pairs were used: ITS5/ITS4 for ITS [26], 5Feur/7CReur for rpb2 [27] and EF1-728F/EF1-1567R or EF1-2218R for tef1 [28,29].

PCRs were carried out in a Bio-Rad C1000 thermocycler (Bio-Rad Laboratories, Hercules, CA, USA) under the following cycling conditions: initial denaturation at 95 °C for 3 min; 35 cycles of denaturation at 95 °C for 30 s, annealing at 55 °C for 35 s, and extension at 72 °C for 1 min; followed by a final extension at 72 °C for 7 min. PCR products were purified and sequenced by Macrogen Inc. (Seoul, Republic of Korea). The resulting twenty-one new sequences were edited and assembled using standard procedures and then deposited in GenBank (Table 1).

### 2.4. Phylogenetic Analysis

Closely related *Trichoderma* species were identified for the strains analyzed in this study using the BLAST algorithm available in NCBI [42]. Reference sequences from type, ex-type, or vouchered strains of related species were obtained from original species descriptions and taxonomic revisions [1,16]. ITS, tef1, and rpb2 sequences of reference strains were retrieved from GenBank and used for phylogenetic comparisons (Table 1). Sequence alignments for each locus (ITS, tef1, and rpb2) were generated using the MAFFT v7.310 online service with default parameters [43]. To ensure consistency and remove ambiguities, the three datasets were concatenated following manual curation using PhyDE v0.9971 (http://www.phyde.de/ (accessed on 7 February 2025)). Pairwise similarities between query and reference properly trimmed sequences were calculated using Clustal Omega (https://www.ebi.ac.uk/Tools/msa/clustalo/ (accessed on 17 July 2025)) following the procedures described by Cai and Druzhinina [1] applying ≥97% identity for tef1 and ≥99% for rpb2 as the thresholds for species-level identification.

Phylogenetic analyses were first performed separately for tef1 and rpb2 to evaluate genealogical concordance under the Genealogical Concordance Phylogenetic Species Recognition (GCPSR) concept [44]. Subsequently, concatenated datasets (tef1+rpb2 and tef1+rpb2+ITS) were analyzed. Phylogenetic reconstructions were conducted using both the Maximum Likelihood (ML) and Bayesian Inference (BI) approaches. ML analyses were carried out with IQ-TREE v3.0.1 [45] under the following best-fit substitution models selected by ModelFinder [46] according to the Bayesian Information Criterion (BIC): HKY + F + G4 for tef1, TN + I + G4 for rpb2, and TN + R2 for the concatenated datasets. Node support values were estimated using 1000 ultrafast bootstrap replicates [47]. BI analyses were performed in MrBayes v3.2.6 [48] under the general time-reversible model (GTR + I + G), with unlinked parameters and rate variation among sites. Two independent runs of four Markov chains were executed for 5 × 10^6^ generations, sampling every 1000 generations, with the first 25% of trees discarded as burn-in. Consensus trees were visualized and edited using FigTree v1.4.4 [49].

## 3. Results

### 3.1. Phylogenetic Analysis

After alignment, the trimmed tef1 and rpb2 dataset consisted of 34 strains with 561 and 813 characters, respectively. The dataset of tef1 + rpb2 and tef1 + rpb2 + ITS consisted of 36 strains with 1735 and 2333 characters, respectively (tef1 = 652, rpb2 = 1083, ITS = 598). The datasets included 23 *Trichoderma* species from the Harzianum and Virens clades, along with two species used as outgroup taxa: *T. chlamydosporum* (Strictipile clade) and *T. ganodermatis* (non-clade assigned). The phylogenetic reconstructions obtained through Bayesian Inference (BI) were consistent with the topology inferred via Maximum Likelihood (ML). Therefore, only the ML trees are presented, incorporating posterior probability (PP) values from the BI analysis. The recovered topology by tef1 + rpb2 analysis is the same by tef1 + rpb2 + ITS analysis; therefore, only the last tree is presented. No substantial changes were observed between the topologies of the tef1 and rpb2 separate analyses concerning the new studied strains (Figure 2). Consistent with the single-locus tef1 and rpb2 analyses, the phylogeny of multi-locus analysis revealed the presence of three *Trichoderma* species amongst the seven strains analyzed in this study (Figure 3). Within the Harzianum clade, *T. endophyticum* (PP/BT = 1/80) and *Trichoderma sanisidroense* (PP/BT = 1/100) were identified, the latter representing a new species clustering with *T. breve*, *T. zelobreve*, *T. peruvianum* and *T. corneum* (PP/BT = 1/99). The third, *Trichoderma jilotepecense*, represents another new species (PP/BT = 1/100) placed within the Virens clade (PP/BT = 1/100), clustering with *T. neocrassum* and *T. virens* (PP/BT = 0.98/82). The phylogenetic placement of these strains was further supported by distinctive cultural and morphological traits, which are described in detail under the Taxonomy section.

### 3.2. Taxonomy

#### 3.2.1. *Trichoderma endophyticum* F.B. Rocha, Samuels & P. Chaverri (Figure 4)

##### Description

Cultural characteristics: colony radius on CMD after 72 h: 36 mm at 25 °C, 53 mm at 30 °C, 11 mm at 35 °C. Colonies hyaline, radial, indistinctly zonate. Aerial hyphae white, abundant at colony margins. Minute pustules present in the center, sparsely distributed, initially white, gradually turning green after 3 days. Colony radius on PDA after 72 h: 51 mm at 25 °C; at 30 °C, aerial mycelium covering the entire plate; 22 mm at 35 °C. On PDA, colonies are cottony, white, with dense mycelium. Aerial hyphae are abundant and cottony. Conidia are produced radially, forming within 48 h, initially at the colony margin, white at first and gradually turning green. Colony radius on SNA after 72 h: 36 mm at 25 °C, 48 mm at 30 °C, 16 mm at 35 °C. Colonies are translucent, with scattered pustules across the plate, initially white, turning dark green after 3 days. A yellow pigment was observed on PDA.

**Figure 4 jof-11-00856-f004:**
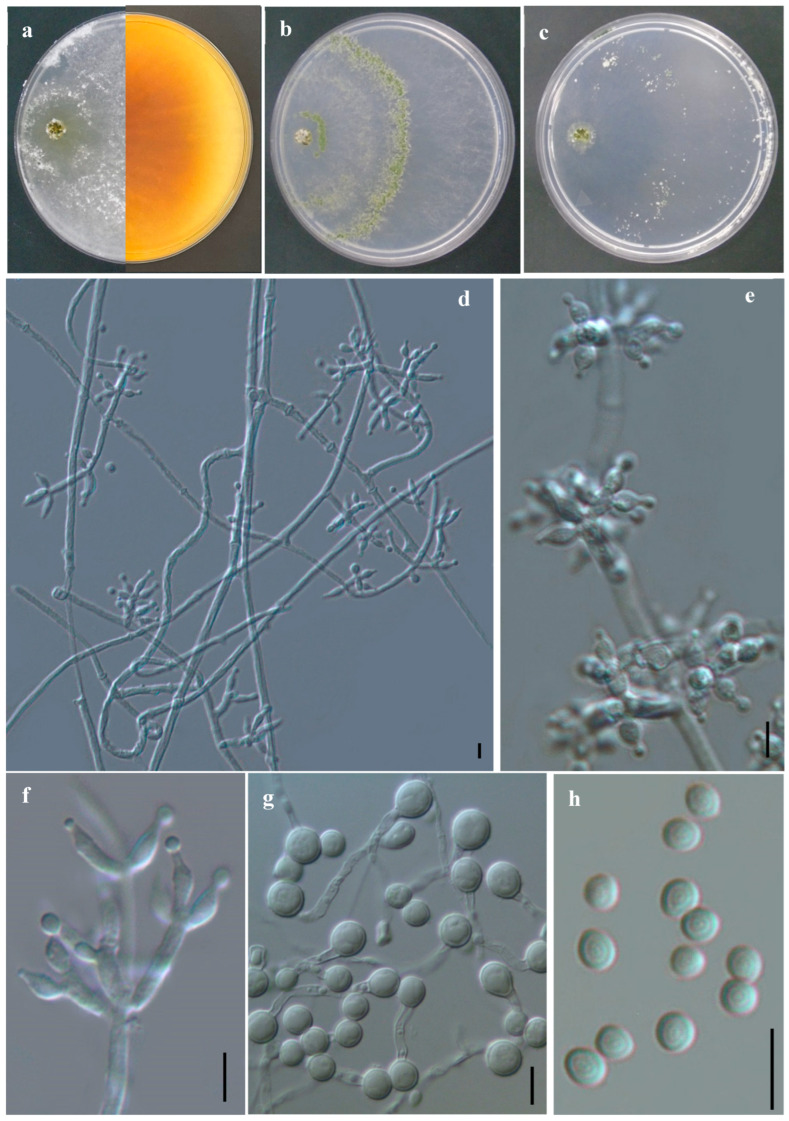
*Trichoderma endophyticum* (IE7006). (**a**–**c**) Cultures after 7 d at 25 °C (**a**) on CMD, (**b**) on PDA, (**c**) on SNA); (**d**–**f**) Conidiophores, phialides and conidia formed on PDA; (**g**) chlamydospores; (**h**) Conidia. Scale bars: 10 µm. Index Fungorum: 809989.

Sexual morph: Unknown. Asexual morph: Conidiophores pyramidal with opposite branches, each ending in a verticillate whorl. Phialides ampulliform, 11.0–15.0 × 2.5–3.0 μm. Conidia globose, subglobose to ovoid, 3.5–5.0 × 4.0–4.5 μm, smooth-walled. Chlamydospores 5.5–9.5 µm.

Material examined: Mexico, Veracruz, Jilotepec, Los Bambus, 19°36′44″ N, 96°60′16.5″ W, 1370 m alt., isolated from coffee plantations soil, 15 August 2021, R. Arias RA48 (IE7006). 

Known distribution: Ecuador and Peru [17], Argentina [50], Brazil [51] and Mexico (this study).

Known hosts and substrates: endophytic on lower stems of *Theobroma gileri* and *Hevea* spp. [17]; reported from cultivated and undisturbed soils, fallen branches, foliar rachises and petioles of *Butia yatay*, palm bases of *B. yatay*, and petioles of *Syagrus romanzoffiana* [50]; isolated from the ascidian *Botrylloides giganteus* [51], and from coffee plantation soils (this study).

Notes: According to the pairwise similarity analysis, rpb2 sequence of IE7006 has high similarity (≥99%) to three species, *T. afarasin* and *T. shaanxiensis* (99.51%, 4 changes/813b) and *T. endophyticum* (99.26%, 6 changes/813b). Tef1 sequences showed five similar (≥97%) species, although *T. endophyticum* (98.28%, 9 changes/523b) and *T. afarasin* (98.28%, 9 changes/523b) were the most similar. The phylogenetical tree based on tef1 and rpb2 separately does not delimit *T. endophyticum* from *T. afarasin*, causing conflict in the identity of the strain IE7006 (Figure 1). However, in the multilocus tree, the strain IE7006 clusters with *T. endophyticum* (Figure 2). The morphological features of this isolate are consistent with the description by Chaverri et al. [17]. The range in conidial dimensions may vary depending on the isolate, although it generally falls within 3.0–6.0 μm.

The original description reported that it was isolated from Ecuador and Peru as an endophyte in tropical trees [17]. In Argentina, it is found on cultivated and undisturbed soils, on fallen branches, petioles and spathes of native palms [50]. In Brazil, it was reported in marine environments [51]. In this study, *T. endophyticum* was isolated from soil in a coffee plantation located in a tropical region of Veracruz and is reported here for the first time in Mexico. This statement was verified through the recent review by Ahedo-Quero et al. [24], which confirms the absence of previous records for the country.

#### 3.2.2. *Trichoderma jilotepecense* R. M. Arias & Gregorio-Cipriano, sp. nov. (Figure 5)

Etymology: The specific epithet “jilotepecense” refers to the locality, Jilotepec where the holotype was found.

**Figure 5 jof-11-00856-f005:**
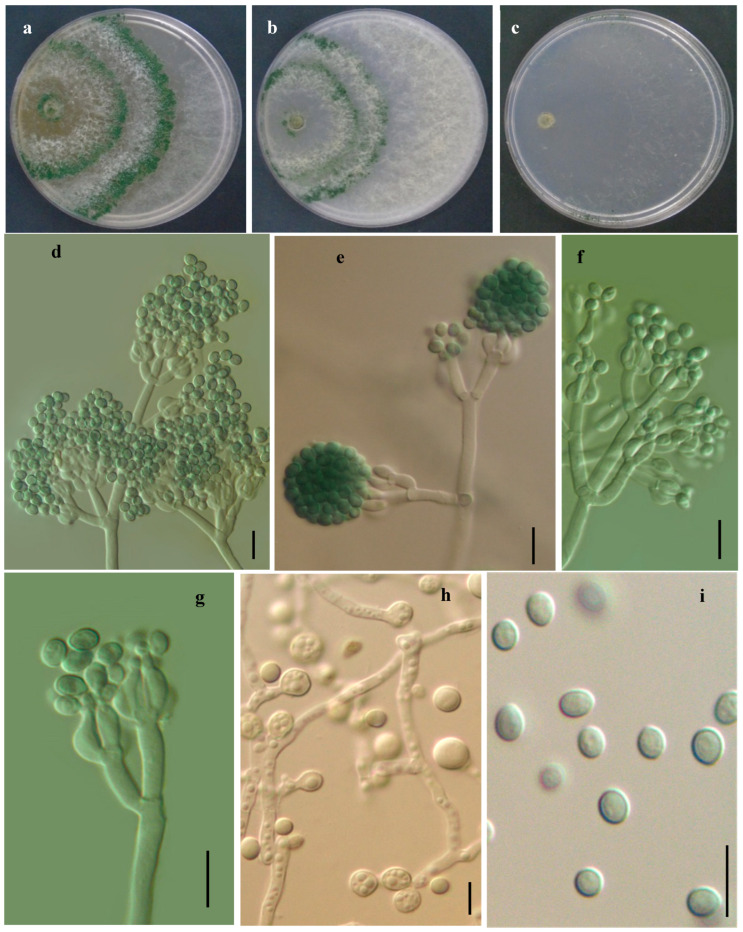
*Trichoderma jilotepecense*. (IE7000 Holotype). (**a**–**c**) Cultures after 7 d at 25 °C (**a**) on CMD, (**b**) on PDA, (**c**) on SNA); (**d**–**g**) Conidiophores, phialides and conidia; (**h**) chlamydospores; (**i**) conidia formed on PDA. Scale bars: 10 µm. Index Fungorum: IF903902.

##### Description

Cultural characteristics: Colony radius on CMD after 72 h: 34 mm at 25 °C, mycelium covering the plate at 30 °C, 22 mm at 35 °C. Abundant, cottony aerial mycelium with radial zonation; conidia abundant and arranged in two or three concentric rings around the inoculation point. Colony radius on PDA after 72 h: 38 mm at 25 °C, 27 mm at 30 °C, 24 mm at 35 °C. Abundant, cottony aerial mycelium with radial zonation; conidia abundant and arranged in two or three concentric rings around the inoculation point. No distinct odor was noted. Colony radius on SNA after 72 h: 34 mm at 25 °C, mycelium covering the plate at 30 °C, 25 mm at 35 °C. Hyaline colonies, slightly greenish at the edges, sparse mycelium. Aerial hyphae scarce.

Sexual morph: Unknown. Asexual morph: Gliocladium-like conidiophores generally with one or two fertile branches, branching irregularly near the tip, penicillium-shaped, sometime unbranched and sterile near the base. Phialides lageniform or ampulliform, 6.0–16.5 × 3.0–4.5 µm. Conidia oval to ellipsoid, less commonly subglobose or oblong, green, smooth, 3.1–4.7 × 3.0–4.7 µm. Chlamydospores abundant, formed in aerial and submerged mycelium, globose to subglobose, terminal and intercalary, hyaline, thick-walled, 4.0–6.5 μm diam.

Material examined: Type: México, Veracruz, Jilotepec, La Barranca, 19°36′40″ N, 96°55′40.5″ W, 1484 m alt., isolated from coffee plantations soil, 15 August 2021, R. Arias RA12 (Holotype IE7000). Additional material: México, Veracruz, Jilotepec, La Barranca, 19°36′37″ N, 96°55′44″ W, 1427 m alt., isolated from coffee plantations soil, 15 August 2021, R. Arias RA24 (IE7001) and R. Arias RA26 (IE7002).

**Notes:** Phylogenetically closets species of *Trichoderma jilotepecense* showed high similarities in rpb2 sequences, *T. neocrassum* (99.14%, 8 changes/813b), followed by *T. crassum* (98.81%, 9 changes/813b), and *T. virens* (98.28% 14 changes/813b). Contrary, no high similar (≥97%) tef1 sequences were detected; *T. virens* was the closest (95.22%, 26 changes/544b), followed by *T. neocrassum* (95.06%, 23 changes/466b), and *T. crassum* (94.67%, 29 changes/544b). In addition to molecular differentiation, *T. jilotepecense* is clearly distinguished by morphological features from *T. virens* [40], *T. crassum* [52] and *T. neocrassum* [36], such as phialide and conidial dimensions and growth on culture media. The phialides of *T. jilotepecense* are notably longer and more variable (6.0–16.5 × 3.0–4.5 µm) compared to those of *T. crassum* (4.4–9.5 × 3.0–4.2 µm); *T. virens* (4.5–10.0 × 2.8–5.5 µm), which may serve as an important diagnostic feature. There is also a slight variation in conidial size. *T. jilotepecense* exhibits smaller conidia (3.1–4.7 × 3.0–4.7 µm) compared to *T. crassum* (3.7–5.3 × 2.6–3.7 µm), *T. virens* (3.5–6.0 × 2.8–4.1 µm) and *T. neocrassum* (5.9–6.4 × 4.7–4.9 µm). Regarding growth in SNA media, *T. jilotepecense* exhibited faster growth compared to *T. virens* and *T. neocrassum*, which may reflect a more efficient adaptation to nutrient-limited conditions. On PDA, all species grew moderately, but *T. jilotepecense* stood out due to its rapid colonization. *T. jilotepecense*, *T. crassum* and *T. neocrassum* grow at 35 °C while *T. virens* does not grow. *T. jilotepecense* was isolated from coffee plantation soil in a tropical region of Mexico, while *T. crassum* was obtained from forest soil in Canada. *T. neocrassum* has been reported from wood and soil samples collected in Belize (North America) and Thailand. In the case of *T. virens*, its isolation has been documented from various substrates, including soil, fallen wood, and other fungi, mainly in temperate regions of the United States.

#### 3.2.3. *Trichoderma sanisidroense* R. M. Arias & Heredia, sp. nov. (Figure 6)

Etymology: The specific epithet “sanisidroense” refers to the locality, San Isidro, Jilotepec where the holotype was found.

**Figure 6 jof-11-00856-f006:**
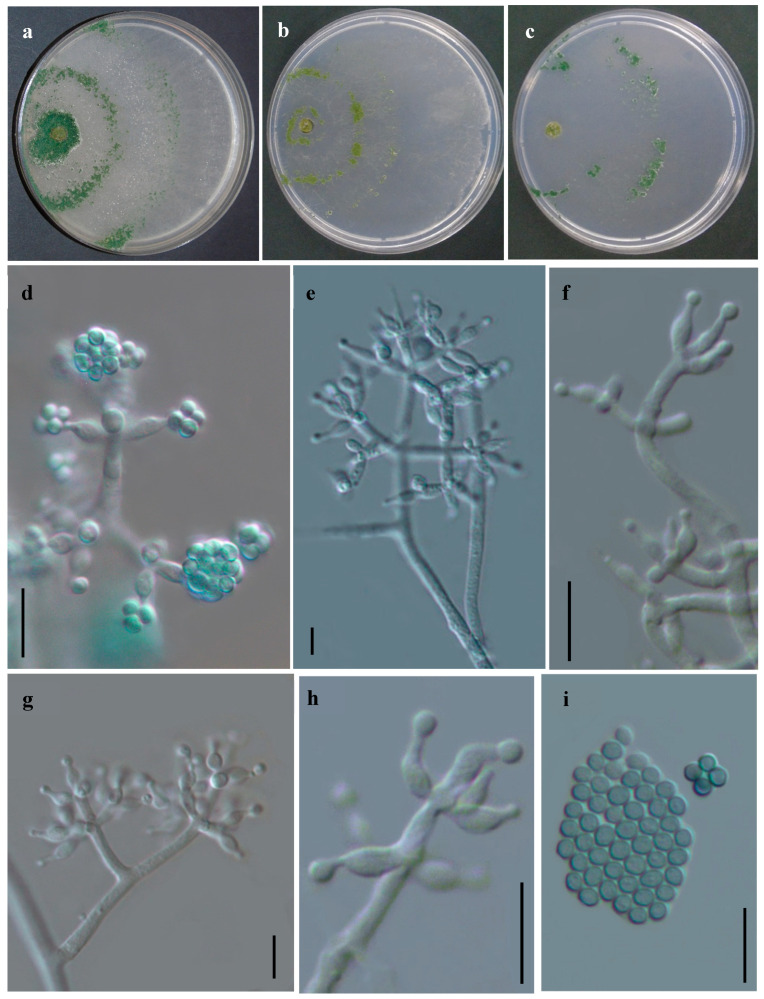
*Trichoderma sanisidroense*. (IE7004 Holotype). (**a**–**c**) Cultures after 7 d at 25 °C (**a**) on CMD, (**b**) on PDA, (**c**) on SNA); (**d**–**h**) Conidiophores, phialides and conidia formed in PDA; (**i**) Conidia. Scale bars: 10 µm. Index Fungorum: IF903901.

##### Description

Cultural characteristics: Colony radius on CMD after 72 h: 42 mm at 25 °C, 41 mm at 30 °C, 15 mm at 35 °C. Aerial mycelium sparse greenish, conidia forming abundantly within 72–96 h from aerial mycelium in broad concentric bands or flat, often coalescing pustules to form a continuous lawn covering extensive areas. Mycelium whitish at the edges. Colony radius on PDA after 72 h: 43 mm at 25 °C, 21 mm at 30 °C, covering the plate after 6 days at 25 °C. On PDA, abundant green aerial mycelium with white base, cottony, radial zoning; conidia abundant and arranged in two or three concentric rings around the point of inoculation. No distinct odor noted. Colony radius on SNA after 72 h: 38 mm at 25 °C, 49 mm at 30 °C, 10 mm at 35 °C. broad concentric rings, first in the aerial mycelium and later in compact, flat, gray-green, 1–3 mm diam. pustules beginning around the inoculum, the degree of pustule formation variable and dependent on the culture, sometimes no pustules observed. No pigment noted, odor indistinct.

Sexual morph: Unknown. Asexual morph: Symmetrical arboreal conidiophores, straight to slightly curved, branches in whorls. The branches are usually perpendicular to the main axis, terminating in 2–3 phialides. Phialides ampulliform to lageniform, emanating in a whorled manner, sometimes arising singly directly from the main axis, 4.5–10.0 × 2.0–3.6 µm. Conidia globose, green, smooth, 2.0–2.6 × 2.0–2.6 µm. Chlamydospores absent.

Material examined: Type: México, Veracruz, Jilotepec, San Isidro, 19°36′13″ N, 96°54′47″ W, 1370 m alt., isolated from coffee plantations soil, 15 August 2021, R. Arias RA55 (Holotype IE7004). Additional material: Ibid. R. Arias RA53 (IE7003).

México, Veracruz, Jilotepec, San Isidro, 19°36′11″ N, 96°54′46″ W, 1412 m alt., isolated from coffee plantations soil, 15 August 2021, R. Arias RA78 (IE7005).

Notes. Between the closest species to *Trichoderma sanisidroense*, *T. breve* was the most similar (98.4%, 13 changes/813b) in rpb2 sequences, followed by *T. zelobreve* (98.28%, 14 changes/813b), *T. corneum* (98.15%, 14 changes/756b), and *T. peruvianum* (97.59%, 18 changes/746b). On the other side, tef1 sequences of *T. breve* (99.39%, 2 changes/327b) and *T. zelobreve* (99.36%, 2 changes/315b) showed high similarities with *T. sanisidroense*. Although, it is important to consider that both lack an initial fragment of 234b or 246b in their sequences, respectively. *Trichoderma corneum* was clearly differentiated (84.73%, with more than 60 changes). While *T. peruvianum* lacks barcode fragments of tef1 and was not possible to compare. *T. sanisidroense* is clearly distinguished by its smaller conidia (2.0–2.6 × 2.0–2.6 µm) and narrower phialides (4.5–10.0 × 2.0–3.6 µm) compared to *T. breve* (conidia 2.5–3.5 × 2.5–3.1 µm; phialides (6.7–10.0 × 2.8–3.9 µm) [31] and *T. peruvianum* (conidia 3.1–4.3 × 2.4–3.7 µm); phialides (6.6–11.4 × 5.8–9.0 µm) [37]. It also differs from *T. zelobreve* [35] which produces wider conidia (4.0–6.0 × 2.6–3.2 µm) and larger phialides (4.0–6.0 × 2.6–3.2 µm). Furthermore, *T. sanisidroense* does not form chlamydospores, in contrast to *T. breve* and *T. peruvianum*, which do produce them.

Colony morphology also provides distinguishing features. Colonies of *T. sanisidroense* are dense, cottony, and light green, whereas those of *T. breve* and *T. zelobreve* are less dense and exhibit colorations ranging from light green to grayish [31,37]. Regarding conidiophore structure, *T. sanisidroense* develops tree-like conidiophores bearing ampulliform to lageniform phialides, while the other species exhibit more irregular or robust conidiophores with phialides of variable morphology. *T. sanisidroense* was isolated from coffee plantation soil in a tropical region of Mexico, whereas *T. breve* and *T. zelobreve* were isolated from soils in China, specifically in Yanqing County and Chaoyang District, Beijing, respectively. The species *T. peruvianum* was also isolated from agroecosystem soil, but in this case, from a cacao cultivation system in Peru.

## 4. Discussion

In Mexico, 57 *Trichoderma* species have been described to date, with the highest diversity recorded in the states of Tabasco (20 species) and Veracruz (18 species). Most studies in the country have focused on the biological control potential of these fungi, while research addressing their diversity and taxonomy remains limited. Previous investigations have relied primarily on morphological criteria or single-locus molecular analyses, typically based on ITS, tef1, or LSU sequences [53,54,55,56,57,58,59]. However, because *Trichoderma* comprises numerous cryptic and closely related taxa, accurate identification now requires multilocus approaches combining at least three barcode regions (ITS, tef1, and rpb2) [1]. The present study addresses this gap by applying an integrative taxonomic framework to characterize *Trichoderma* species from coffee plantation soils in Veracruz, thereby contributing to a more comprehensive understanding of their diversity in Mexico.

To date, 29 *Trichoderma* species have been reported from coffee rhizospheres [23,60,61,62,63]. In this study, two new species (*T. sanisidroense* and *T. jilotepecense*) are described, and *T. endophyticum* is recorded for the first time in Mexico. The integration of phylogenetic, morphological, ecological, and biogeographic evidence enabled reliable species delimitation within the genus, confirming the diagnostic utility of rpb2 and tef1 as key loci for *Trichoderma* taxonomy. The multilocus phylogenetic framework employed (ITS, rpb2, tef1), together with detailed morphological comparisons, provided robust support for the recognition of these new taxa.

Phylogenetic analyses revealed that *T. sanisidroense* belongs to the Harzianum clade, whereas *T. jilotepecense* is part of the Virens clade. Cultures of *T. sanisidroense* exhibited the formation of pustules, consistent with observations by Ye et al. [21], who reported considerable variation in this trait among members of the Harzianum clade. Similarly, the absence of chlamydospores in *T. sanisidroense* agrees with previous reports describing their rarity within this lineage. *T. jilotepecense*, a member of the *Virens* clade, is characterized by green conidia, rapid growth, and *Gliocladium*-like conidiophores. In Mexico, two other members of this clade, *T. virens* and *T. crassum*, have previously been identified based on morphological and limited molecular data [55,64]. However, as demonstrated by Cai and Druzhinina [1], ITS sequences alone are insufficient for precise species delimitation, underscoring the importance of multilocus phylogenetic frameworks such as the one applied here.

The identification of *T. sanisidroense*, *T. jilotepecense*, and *T. endophyticum* from coffee soils reveals the remarkable yet insufficiently documented fungal diversity inhabiting Mexico’s tropical agroecosystems. Shade-grown coffee plantations, characterized by stable microclimatic conditions and organic matter–rich soils, provide complex microhabitats that promote the diversification and persistence of saprotrophic and mycoparasitic fungi such as *Trichoderma*. From a biogeographic perspective, these findings suggest that Mesoamerica may represent a secondary diversification center for the genus, where ecological heterogeneity and long-term agroforestry practices have fostered niche specialization and local adaptation.

Beyond their taxonomic relevance, the newly identified species expand the known ecological and biotechnological potential of *Trichoderma*. Their occurrence in traditionally managed coffee systems underscores the adaptive capacity of the genus to persist and evolve under moderate anthropogenic influence. Such agroecosystems thus serve not only as reservoirs of fungal biodiversity but also as evolutionary arenas that facilitate the emergence of novel lineages.

Given their adaptation to coffee soils, the new species described here represent promising candidates for future studies on biocontrol and biofertilization. Their potential functional attributes—such as nutrient solubilization, pathogen suppression, and stress tolerance—warrant further investigation to assess their applicability in sustainable coffee production and broader agroecological contexts.

Overall, this study contributes to a deeper understanding of *Trichoderma* diversity in Mexico and highlights the value of integrative taxonomy for uncovering hidden fungal lineages in tropical soils. The results demonstrate that traditional coffee agroecosystems harbor unrecognized fungal diversity and emphasize the need for continued research combining morphology, multilocus phylogenetics, and ecological data to elucidate the evolutionary dynamics of *Trichoderma* in tropical environments. Such integrative approaches are essential to reveal cryptic diversity and to link fungal systematics with ecosystem sustainability.

## 5. Conclusions

This study describes two new species, *Trichoderma sanisidroense* and *T. jilotepecense*, and reports *T. endophyticum* for the first time in Mexico. An integrative approach combining multilocus phylogenetic analyses (ITS, tef1, rpb2) with detailed morphological characterization provided strong evidence supporting their taxonomic distinction within the Harzianum and Virens clades. These findings demonstrate that traditionally managed coffee soils in Veracruz harbor previously unexplored fungal diversity and underscore their ecological importance as reservoirs of beneficial *Trichoderma* species with potential applications in sustainable agriculture. Future research should focus on elucidating their ecological roles and evaluating their biotechnological potential within coffee agroecosystems.

## Figures and Tables

**Figure 1 jof-11-00856-f001:**
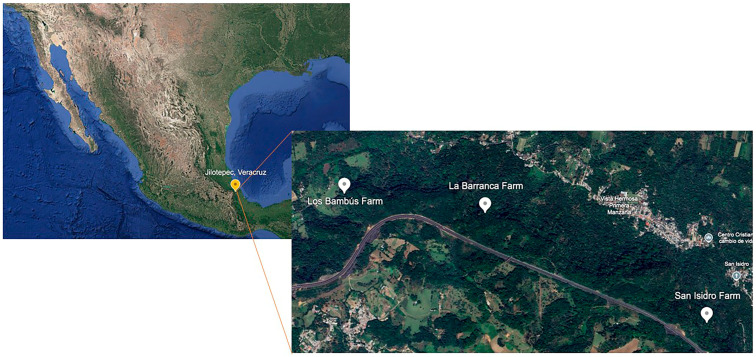
Location of Coffee Farms in Jilotepec, Veracruz, Mexico.

**Figure 2 jof-11-00856-f002:**
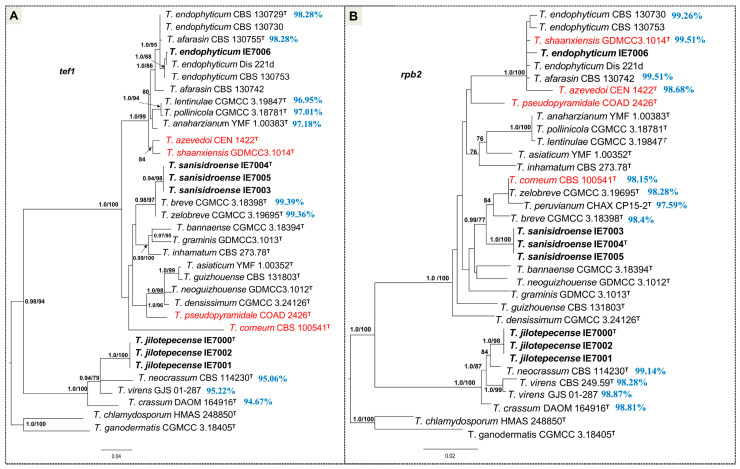
Phylogenetic tree generated by Maximum Likelihood by separate tef1 (**A**) and rpb2 (**B**) sequences from Harzianum and Virens clades of *Trichoderma*. Bayesian posterior probability (≥0.90) and Maximum Likelihood bootstrap support values (≥75%) are indicated at the nodes. T = Type specimen. *Trichoderma chlamydosporum* and *T. ganodermatis* were used as outgroups. The percentage of pairwise similarities between the studied strains (highlighted in bold) and their closest species is indicated in front of the reference strain in blue. Species that do not share the same position in both phylogenies are in red.

**Figure 3 jof-11-00856-f003:**
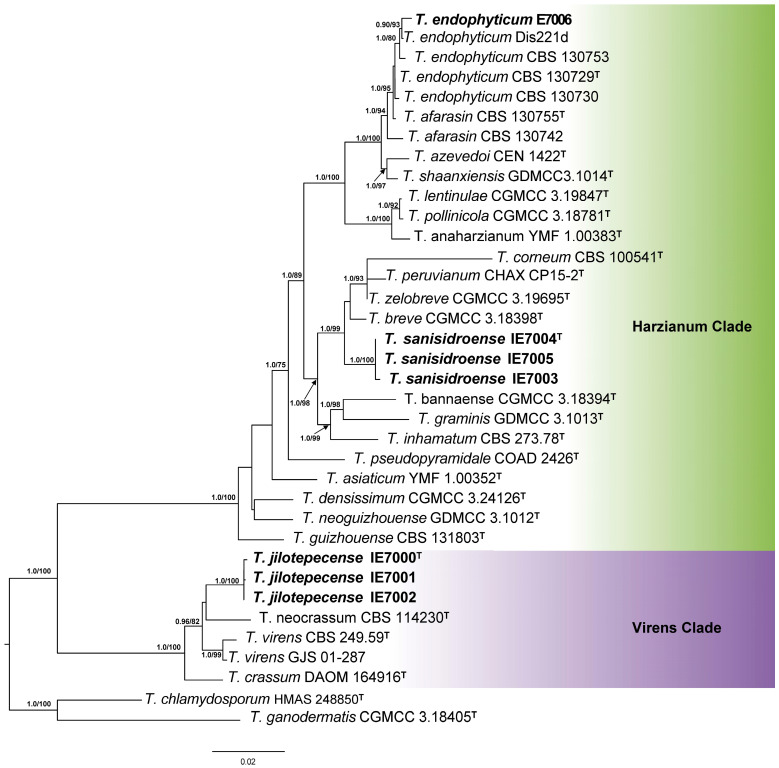
Phylogenetic tree generated by Maximum Likelihood, based on concatenated sequences of tef1, rpb2 and ITS loci from the Harzianum and Virens clades of *Trichoderma*. Bayesian posterior probability (≥0.90) and Maximum Likelihood bootstrap support values (≥75%) are indicated at the nodes. The strains reported in this study are highlighted in bold. T = Type specimen. *Trichoderma chlamydosporum* and *T. ganodermatis* were used as outgroups. This tree represents only the most closely related species of the specimens; however, it was recovered a phylogenetic reconstruction including all the species of the Harzianum clade with all the available reference material in GenBank (until May 2025), it can be consulted in the Appendix A.

**Table 1 jof-11-00856-t001:** Information of sequences used for molecular phylogenetic analyses.

Species	Strain/Voucher	Origin	GenBank Accession Number			References
			ITS	*tef1*	*rpb2*	
*Trichoderma afarasin*	G.J.S. 06-98 = CBS 130501	Cameroon	FJ463400	FJ463327	—	[17]
*T. afarasin*	G.J.S. 99-227 = CBS 130755 **^T^**	Cameroon	AY027784	AF348093	—	[17]
*T. anaharzianum*	YMF 1.00383 **^T^**	China	MH113931	MH183182	MH158995	[18]
*T. asiaticum*	YMF 1.00352 **^T^**	China	MH113930	MH183183	MH158994	[18]
*T. azevedoi*	CEN 1422 **^T^**	Brazil	MK714902	MK696660	MK696821	[30]
*T. bannaense*	HMAS 248840 = CGMCC 3.18394 **^T^**	China	KY687923	KY688037	KY687979	[31]
*T. breve*	HMAS 248844 = CGMCC 3.18398 **^T^**	China	KY687927	KY688045	KY687983	[31]
*T. corneum*	G.J.S. 97-75 = CBS 100541 **^T^**	Thailand	—	AY937431	KJ842183	[32,33]
*T. crassum*	DAOM 164916 **^T^**	Canada	EU280067	EU280048	KJ842185	[32]
*T. densissimum*	T32434 = CGMCC 3.24126 **^T^**	China	—	OP357971	OP357966	[22]
*T. endophyticum*	Dis 217h = CBS 130730	Ecuador	FJ442242	FJ463314	FJ442721	[17]
*T. endophyticum*	Dis 217a = CBS 130729 **^T^**	Ecuador	FJ442243	FJ463319	—	[17]
*T. endophyticum*	Dis 218f = CBS 130753	Ecuador	FJ442246	FJ463326	FJ442722	[17]
*T. endophyticum*	Dis 221d	Ecuador	FJ442248	FJ463389	FJ442794	[17]
** *T. endophyticum* **	**IE7006**	**Mexico**	**PV687649**	**PV694681**	**PV694674**	**This study**
*T. graminis*	YNE00410 = GDMCC3.1013 **^T^**	China	—	OR779514	OR779491	[5]
*T. guizhouense*	HGUP0038 = CBS 131803 **^T^**	China	JN191311	JN215484	JQ901400	[34]
*T. inhamatum*	CBS 273.78 **^T^**	Colombia	FJ442680	AF348099	FJ442725	[16]
** *T. jilotepecense* **	**IE7000 ^T^**	**Mexico**	**PV687646**	**PV694678**	**PV694671**	**This study**
** *T. jilotepecense* **	**IE7001**	**Mexico**	**PV687647**	**PV694679**	**PV694672**	**This study**
** *T. jilotepecense* **	**IE7002**	**Mexico**	**PV687648**	**PV694680**	**PV694673**	**This study**
*T. lentinulae*	HMAS 248256 = CGMCC 3.19847 **^T^**	China	MN594469	MN605878	MN605867	[35]
*T. neocrassum*	G.J.S. 01-227 = CBS 114230 **^T^**	Thailand	—	JN133572	AY481587	[16,36]
*T. neoguizhouense*	T33324 = GDMCC3.1012 **^T^**	China	—	OR779516	OR779487	[5]
*T. peruvianum*	CHAX CP15-2 **^T^**	Peru	—	MW480145	MW480153	[37]
*T. pollinicola*	LC11682 = LF1542 = CGMCC 3.18781 **^T^**	China	MF939592	MF939619	MF939604	[38]
*T. pseudopyramidale*	COAD 2426 **^T^**	Ethiopia	—	MK044131	MK044224	[39]
** *T. sanisidroense* **	**IE7003**	**Mexico**	**PV687650**	**PV694682**	**PV694675**	**This study**
** *T. sanisidroense* **	**IE7004 ^T^**	**Mexico**	**PV687651**	**PV694683**	**PV694676**	**This study**
** *T. sanisidroense* **	**IE7005**	**Mexico**	**PV687652**	**PV694684**	**PV694677**	**This study**
*T. shaanxiense*	T32000 = GDMCC3.1014 **^T^**	China	—	OR779513	OR779486	[5]
*T. virens*	Gli 39 = ATCC 13213 = CBS 249.59 **^T^**	USA	AF099005	AF534631	AF545558	[16,36,40]
*T. virens*	G.J.S. 01-287	Cote d’Ivoire	DQ083023	AY750894	EU341804	[41]
*T. zelobreve*	HMAS 248254 = CGMCC 3.19695 **^T^**	China	MN594474	MN605883	MN605872	[35]
*T. chlamydosporum* **^OT^**	HMAS 248850 = CGMCC 3.18401 **^T^**	China	KY687933	KY688052	KY687989	[31]
*T. ganodermatis* **^OT^**	HMAS 248856 = CGMCC 3.18405 **^T^**	China	KY687939	KY688060	KY687995	[31]

Letters in bold indicate the sequences generated in this study. **^OT^** = Out group; **^T^** = Type specimen.

## Data Availability

All sequence data are available in NCBI GenBank following the accession numbers in the manuscript.

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
