# Peer review of "Integrative Taxonomy Reveals Two New Trichoderma Species and a First Mexican Record from Coffee Soils in Veracruz"

_jof, 2025, doi:10.3390/jof11120856_

Round 1
Reviewer 1 Report
Authors detail on two new Trichoderma species and on a known species reported for the first time in Mexico in soil samples collected in three different coffee orchards supposed as representative of the agroecosystems in Veracruz. So additional information on the composition of Trichoderma genus are herein detailed.
In the study, two new species, Trichoderma jilotepecense and Trichoderma sanisidroense, are reported. Additionally, the Trichoderma endophyticum was reported as the first record for Mexico. All the three species are well described with respect to their morphological aspects and micromorphological features. No information on their distribution is supplied so the reader can understand if these new species are well represented and established in the inspected sites or represent just a unit. Additionally, I cannot understand why the authors detail on taxon of the known specie Trichoderma endophyticum is re-described. The Authors find such differences in the previous description? No information on the other Trichoderma species resident in the inspected sites is reported as well as on the other coexistent species. Important integration on these aspects must be done. Additionally some suggestion are detailed in the attach files.

Author Response
We sincerely appreciate the reviewer’s valuable comments and constructive observations, which have significantly contributed to improving the quality of the manuscript. In response to these suggestions, we have incorporated additional information in the revised version specifying the precise geographic points where the strains were isolated. This inclusion provides a clearer representation of the distribution of Trichoderma species across the three studied sites, complementing the taxonomic description and offering readers a better understanding of the ecological context of the isolates.
Regarding Trichoderma endophyticum, we confirm that the morphological characteristics of our isolate are fully consistent with the original description by Chaverri et al. [17]. Minor variations were observed in the conidial size range, which—as commonly reported for this genus—may vary slightly among isolates but remain within the previously established range (3.0–6.0 μm).
It is also important to clarify that these isolates were previously mentioned in Arias et al. (2021), where more detailed information was provided on ecological aspects, isolation conditions, and the agroecological context of the sampling sites. In that earlier study, the isolates were identified only at the genus level. In the present manuscript, we extend this previous work by applying an integrative taxonomic approach that allows the formal description of two new species and confirms the first record of T. endophyticum for Mexico, based on both morphological and multilocus analyses.
This new contribution therefore complements and expands the existing information, providing robust taxonomic evidence supporting the delimitation of these taxa and their representation within the coffee agroecosystems of Veracruz.
Reviewer 2 Report
1. What is the main question addressed by the research?
This article presents compelling evidence for the existence of two new Trichoderma fungal species sampled in coffee plantation soils of Mexican region Veracruz, T. jilotepecense and T. sanisidroense,and one new species for Mexican soils, T. endophyticum of the same genus.
2. Do you consider the topic original or relevant to the field? Does it address a specific gap in the field? Please also explain why this is/ is not the case. • What does it add to the subject area compared with other published material?
These species are new and have not previously been clearly described, as previous classifications relied primarily on morphological characteristics and lacked molecular biological data. This is why the presented study is original.
3. What specific improvements should the authors consider regarding the methodology?
The authors skillfully used molecular genetic approaches based on three regions increasing the accuracy of determination, namely the internal transcribed spacer (ITS) region, the second largest subunit of RNA polymerase II (rpb2), and the translation elongation factor 1-alpha (tef1), combined with the fungi cultural and morphology characteristics to analyze fungal community samples obtained from coffee plantations.
4. Are the conclusions consistent with the evidence and arguments presented and do they address the main question posed? Please also explain why this is/is not the case.
The research provides valid conclusions that are supported by (i) sufficient data including the updated database data on sequences dated May 2025 used for molecular phylogenetic analyses of reference sequences from type, ex-type, or vouchered strains of known species within the Harzianum and Virens clades of Trichoderma, as well as by (ii) the GenBank accession numbers of the species newly described in the presented research.
5. Are the references appropriate?
The references used are appropriate.
6. Any additional comments on the tables and figures.
The article is accompanied by the excellent illustrations, providing data on geographical locations of sampling and the microscopy insight into the biology of the newly described species. The manuscript is well written and presented in all the necessary detail. Below, I have included a small addition that the authors may use to improve the caption to Table 1.
Methods:
Lines 134-135: Please add a phrase in the title that explains what the bold letters in the table body mean.
Author Response
We sincerely appreciate the reviewer’s positive evaluation and constructive comments. Following their suggestion, we made the corresponding modification to the title of Table 1, adding an explanation of the meaning of the bold letters used in the table body. This improvement contributes to a clearer and more precise presentation of the information.
Reviewer 3 Report
The publication submitted for review is formally correct.
However, basing the entire publication on PCR reactions and the description of strains as if from the key to microorganisms is probably a bit too little, for the current times.
In this form, the publication is not suitable for publication.
No specific comments
Author Response
We sincerely appreciate the time dedicated to reviewing our manuscript and the comments provided. Respectfully, we do not fully agree with the assessment that the study relies solely on PCR reactions and basic morphological descriptions. While the work includes detailed strain descriptions, these are supported by an integrative taxonomic approach that combines morphological, cultural, and molecular evidence.
Multilocus analyses were performed using three gene regions widely recognized for their phylogenetic value in Trichoderma (ITS, tef1, and rpb2), allowing for a robust and up-to-date species delimitation consistent with contemporary taxonomic standards (Chaverri et al., 2015; Bissett et al., 2021). This integrative approach is currently accepted by the scientific community for the description of new filamentous fungal species and complies with international guidelines for fungal taxonomy.
Therefore, we consider that the manuscript goes beyond traditional descriptions and represents a solid and current contribution to the understanding of Trichoderma diversity in coffee soil ecosystems of Mexico, combining molecular and morphological tools in a complementary manner.
Reviewer 4 Report
Dear authors,
Now I have completed reviewing the manuscript "Integrative taxonomy reveals two new Trichoderma species and a first Mexican record from coffee soils in Veracruz". In general, this manuscript is well organized. This study, which focuses on identifying new/known soil fungi, identifies the following areas that require clarification and improvement before the manuscript can be considered for publication. Major comments and suggestions for improvement are in the attached file.
Best regards
Some parts of the manuscript require clarification and improvement before it can be considered for publication

Author Response
We sincerely thank you for the time and attention dedicated to the detailed review of our manuscript, as well as for your thoughtful observations and suggestions to improve its quality. We have carefully reviewed all the comments provided in the attached file and made the corresponding corrections in the text.
We believe that these modifications have strengthened the structure, clarity, and coherence of the manuscript, contributing to a clearer presentation of the results and their taxonomic relevance. We deeply appreciate your valuable input, which has been very helpful in improving our work.
Reviewer 5 Report
Dear editor, thank you for your invitation to review this manuscript entitled ‘Integrative taxonomy reveals two new Trichoderma species and a first Mexican record from coffee soils in Veracruz’. The manuscript includes the study about two species of Trichoderma as a new report from Mexico that he isolated from coffee rhizosphere soils in Veracruz. The study is associated with taxonomy including the cultural and micromorphological characterization with multilocus phylogenetic analyses. The work is timely and contributes meaningfully to fungal biodiversity knowledge in tropical agroecosystems, highlighting Trichoderma’s biotechnological potential. The writing is clear, methods are reproducible, and the structure follows standard taxonomic guidelines. Minor revisions are needed for completeness to enhance phylogenetic depth.
The author mentioned in manuscript about molecular analyses but did not provide any GenBank accessions. I suggest the author to kindly provide these accession numbers before acceptance
The author should provide the Holotype deposition and ex-type culture details that should be included in the taxonomic treatment section.
The author should provide the phylogenetic details and should make sure that tree figures show bootstrap/ML support values ≥70% at key nodes.
The author should include the morphological descriptions by comparing the new species to closest relatives.
The author should explain for T. endophyticum, that why Mexican reports were missed and confirm absence via recent reviews.
Author should make sure that GPS coordinates are provided in text for precision and add a phylogenetic tree with strain labels highlighted.
The author should remove all typing errors and should use the standard for temperature throughout the manuscript.
Recommendation:
Accept after minor revisions.
Dear editor, thank you for your invitation to review this manuscript entitled ‘Integrative taxonomy reveals two new Trichoderma species and a first Mexican record from coffee soils in Veracruz’. The manuscript includes the study about two species of Trichoderma as a new report from Mexico that he isolated from coffee rhizosphere soils in Veracruz. The study is associated with taxonomy including the cultural and micromorphological characterization with multilocus phylogenetic analyses. The work is timely and contributes meaningfully to fungal biodiversity knowledge in tropical agroecosystems, highlighting Trichoderma’s biotechnological potential. The writing is clear, methods are reproducible, and the structure follows standard taxonomic guidelines. Minor revisions are needed for completeness to enhance phylogenetic depth.
The author mentioned in manuscript about molecular analyses but did not provide any GenBank accessions. I suggest the author to kindly provide these accession numbers before acceptance
The author should provide the Holotype deposition and ex-type culture details that should be included in the taxonomic treatment section.
The author should provide the phylogenetic details and should make sure that tree figures show bootstrap/ML support values ≥70% at key nodes.
The author should include the morphological descriptions by comparing the new species to closest relatives.
The author should explain for T. endophyticum, that why Mexican reports were missed and confirm absence via recent reviews.
Author should make sure that GPS coordinates are provided in text for precision and add a phylogenetic tree with strain labels highlighted.
The author should remove all typing errors and should use the standard for temperature throughout the manuscript.
Recommendation:
Accept after minor revisions.
Author Response
We sincerely appreciate your positive comments and valuable suggestions, all of which were carefully addressed in the revised version of the manuscript. Below, we provide a point-by-point response to each of the issues raised:
- GenBank accession numbers: Accession numbers for all sequences used in this study have been incorporated. This information is available in Table 1 (fourth column) within the Methodology section.
- Holotype and ex-type culture deposition: Complete deposition details for the holotype and ex-type cultures have been added in the Material Examined section, following international taxonomic standards.
- Comparison with closely related species: For each new species, detailed morphological comparisons—both microscopic and cultural—were included with their closest phylogenetic relatives. These comparisons are presented in the Results section under Notes.
- Record of T. endophyticum in Mexico: It was clarified that this species had not been previously reported in the country. This statement was verified through the recent review by Ahedo-Quero et al. (2024), which confirms the absence of prior records for Mexico. This information was incorporated into the Results section, also under Notes.
- GPS coordinates: Precise geographic coordinates for each isolated strain were added to provide a more complete and accurate spatial context.
- Correction of typographical errors and temperature standardization: The manuscript was carefully revised to correct minor typographical errors, and the temperature format was standardized throughout the text.
We thank you again for your constructive feedback, which has significantly improved the clarity, accuracy, and overall quality of the manuscript.
Round 2
Reviewer 3 Report
The corrections have improved the publication, but I still think that the description of hyphae fungi alone is not enough. I wish the authors good luck in their further research.
And I leave the decision to the editor.
ok
Author Response
We sincerely appreciate your comment. We value the observation regarding the need for broader ecological or functional information; however, we would like to emphasize that rigorous identification of fungal diversity, including the accurate recognition and delimitation of species, is a fundamental step for understanding their ecological roles, evolutionary relationships, and potential applications.
In this context, our study provides novel data on tropical soil fungi associated with coffee agroecosystems, a productive system of high ecological and economic relevance in Mexico, but still poorly explored in terms of its mycobiota. We consider that building a solid foundation on fungal taxonomy and diversity is essential for subsequent research to address functional, ecological, and biotechnological aspects with greater certainty.
We also respectfully note that the manuscript was submitted to the “Fungal Evolution, Biodiversity and Systematics” section, within the Special Issue “Fungal Diversity in the Americas,” precisely because its main objective is to contribute to the knowledge of fungal diversity in a region underrepresented in current inventories.
We are grateful once again for the reviewer’s good wishes and observations, which we appreciate as an opportunity to strengthen the value and scope of this work. We remain attentive to any further considerations from the editor.
Reviewer 4 Report
Dear Authors,
I have one part that needs you to revise. Regarding the known host and distribution of known species, you need to list which hosts and where this fungi has been found.
For example:
Known distribution: Italy (citation), Mexico (this study), Spain (citation), Ukraine (citation).
Known hosts: from soil (citation; this study), unknown wood (citation).
Thank you
Dear Authors,
I have one part that needs you to revise. Regarding the known host and distribution of known species, you need to list which hosts and where this fungi has been found.
For example:
Known distribution: Italy (citation), Mexico (this study), Spain (citation), Ukraine (citation).
Known hosts: from soil (citation; this study), unknown wood (citation).
Thank you
Author Response
Thank you very much for your accurate and insightful comment. We fully agree that the original sentence required greater clarity and precision. Following your suggestion, we revised the text to improve its readability, avoid ambiguity, and provide a clearer description of the reported hosts.
We sincerely appreciate your observation, as it significantly contributes to strengthening the quality and presentation of the manuscript.
Round 3
Reviewer 3 Report
The revisions have enhanced the publication and I wish the authors much success in their continued research.
ok